# Zinc Starvation Induces Cell Wall Remodeling and Activates the Antioxidant Defense System in *Fonsecaea pedrosoi*

**DOI:** 10.3390/jof10020118

**Published:** 2024-01-31

**Authors:** Tayná Aparecida de Oliveira Santos, Lucas Weba Soares, Lucas Nojosa Oliveira, Dayane Moraes, Millena Silva Mendes, Célia Maria de Almeida Soares, Alexandre Melo Bailão, Mirelle Garcia Silva Bailão

**Affiliations:** 1Laboratory of Molecular Biology, Institute of Biological Sciences, Federal University of Goiás, Goiânia 74690-900, GO, Brazil; 2Department of Microbial Pathogenesis, Yale University School of Medicine, New Haven, CT 06520, USA

**Keywords:** chromoblastomycosis, nutritional immunity, black fungi, zinc transporters

## Abstract

The survival of pathogenic fungi in the host after invasion depends on their ability to obtain nutrients, which include the transition metal zinc. This essential micronutrient is required to maintain the structure and function of various proteins and, therefore, plays a critical role in various biological processes. The host’s nutritional immunity limits the availability of zinc to pathogenic fungi mainly by the action of calprotectin, a component of neutrophil extracellular traps. Here we investigated the adaptive responses of *Fonsecaea pedrosoi* to zinc-limiting conditions. This black fungus is the main etiological agent of chromoblastomycosis, a chronic neglected tropical disease that affects subcutaneous tissues. Following exposure to a zinc-limited environment, *F. pedrosoi* induces a high-affinity zinc uptake machinery, composed of zinc transporters and the zincophore Pra1. A proteomic approach was used to define proteins regulated by zinc deprivation. Cell wall remodeling, changes in neutral lipids homeostasis, and activation of the antioxidant system were the main strategies for survival in the hostile environment. Furthermore, the downregulation of enzymes required for sulfate assimilation was evident. Together, the adaptive responses allow fungal growth and development and reveals molecules that may be related to fungal persistence in the host.

## 1. Introduction

Zinc is an essential micronutrient required to maintain the structure and function of several proteins, such as enzymes and transcription factors, and, therefore, plays a critical role in several biological processes [1]. During an infection, zinc levels in the host are readjusted at systemic, local, and intracellular levels in order to restrict the pathogen’s access to this metal. Zinc restriction to extracellular pathogenic fungi is mediated mainly by calprotectin, a component of neutrophil extracellular traps (NETs) that bind the metal with high affinity [2,3]. Calprotectin and NETs efficiently inhibit the growth of the fungal pathogens *Candida albicans*, *Aspergillus fumigatus*, and *Cryptococcus neoformans* [4,5,6]. Likewise, other proteins of the S100 family, such as psoriasin, a protein present in the skin, has antimicrobial properties as it functions as a zinc chelator [7]. Interestingly, vitamin A deficiency predisposes the host to fungal infections since its derivatives have a fungistatic effect [8]. Zinc deprivation also targets fungal pathogens within macrophages, as demonstrated in *C. albicans* and *Histoplasma capsulatum* [9,10].

Despite the restrictive conditions imposed during an infection, pathogenic fungi are able to obtain zinc and proliferate in the host tissues. A set of membrane transporters, located on the cell surface and in organelles, allows both metal uptake and storage. The expression of these molecules is regulated by orthologs of a zinc-responsive transcription factor that induce the uptake of the metal in response to its low availability [11,12,13]. In some fungi, the expression of transporter genes is also regulated by pH [14].

Zinc transporters belong to two major groups of proteins: the Cation Diffusion Facilitator (CDF) and the Zrt- and Irt-like Protein (ZIP) families. ZIP proteins transport zinc from the extracellular environment or from intracellular membrane compartments into the cytosol, while CDF transporters move zinc out of cytosol into organelles [15,16]. Zrt1 and Zrt2 are, respectively, high- and low-affinity plasma membrane ZIP transporters in *Saccharomyces cerevisiae*. Following uptake, zinc can be transported into the vacuole by the CDF transporter Zrc1. In yeast, zinc assimilation is not pH dependent as it is in *A. fumigatus*. In this mold, the ZIP transporters ZrfA and ZrfB capture zinc in acidic conditions, while ZrfC is responsible for zinc uptake in neutral–alkaline zinc-deprived environments. Zinc assimilation in *C. albicans* also relies on regulation according to pH. Zinc assimilation in neutral–alkaline conditions is mediated by a zincophore system, composed of Pra1 and Zrt1. Pra1 is a secreted protein that binds zinc with high affinity and delivers the ion to the ZIP transporter Zrt1 located at the plasma membrane. Zrt2 mediates zinc uptake mainly in acidic conditions and is recognized as the principal zinc importer in *C. albicans* [13].

The deletion of plasma membrane zinc transporter genes in *A. fumigatus* [17] and *Cryptococcus* species [18,19] as well as the silencing of a zinc importer in *H. capsulatum* [20] have revealed that such molecules are essential in the pathogenesis of mycoses caused by these microorganisms. Additionally, the lack of Pra1 in *C. albicans* compromises the invasion of endothelial cells [21], and the loss of Pra1 and Zrt1 in *Blastomyces dermatitidis* impairs the establishment of disease in mice infected intratracheally [22], demonstrating the importance of zinc acquisition in different sites of infection.

The dematiaceous fungus *Fonsecaea pedrosoi* is the main etiological agent of chromoblastomycosis (CBM), a chronic infection of the skin and subcutaneous tissues that is distributed worldwide [23,24]. Together with sporotrichosis, CBM is the main implantation mycosis recognized by the World Health Organization as a fungal neglected tropical disease [25,26]. CBM treatment is a challenge since the disease presents a recalcitrant nature. No standard treatment is available, and therapy includes chemotherapy with antifungal drugs, physical methods applied directly to the lesions, or an association of both [27]. Lesions initiate with a traumatic inoculation of hyphae or conidia, which transform into muriform cells in tissues. During the infection process, hyphae are also found in the lesions, since muriform cells can germinate [25,28,29]. Initially, lesions develop as a papule that can further evolve into plaque, nodule, verrucous, or cicatricial appearance [27,30]. CBM lesions present a characteristic granulomatous reaction containing inflammatory infiltrates rich in neutrophils and macrophages forming multinucleated giant cells [31,32]. NETs released by neutrophils challenged with *F. pedrosoi* hyphae can kill fungal cells [33]. Since NETs contain calprotectin, *F. pedrosoi* probably undergoes zinc starvation during the infection process.

An in silico analysis of the *F. pedrosoi* genome conducted previously has demonstrated the presence of ZIP and CDF transporters involved in zinc uptake and storage as well as the secreted chelator protein Pra1 [34]. Here we demonstrated the regulation of these genes by zinc at the transcriptional level. In addition, the adaptive responses to zinc deprivation were investigated by a proteomic approach. Cell wall structure, oxidative stress status, and sulfur assimilation were the main pathways impacted by zinc availability.

## 2. Material and Methods

### 2.1. Strains and Growth Conditions

The *F. pedrosoi* CBS 271.37 strain (ATCC 18658) was used in all the experiments. The fungus was maintained in a Potato Dextrose Agar medium at 28 °C for 7 days. Conidia were obtained by inoculation of mycelia fragments into Potato Dextrose Broth (20% potato infusion, 2% glucose), pH 5.6, under agitation at 150 rpm, 28 °C, for 7 days. The culture was vortexed for 1 min to release the conidia, filtered through sterile glass wool, and centrifuged at 3000× *g* for 5 min. The conidia were washed twice and resuspended in phosphate buffered saline (PBS).

To analyze growth in different concentrations of zinc, the chemically defined MMcM medium [35], pH 6.8, glucose 1%, was used. For this, conidia were counted in Neubauer’s chamber to adjust the density to 10^8^. Then, 10^6^ to 10^3^ conidia were plated on MMcM agar plates containing different concentrations of zinc (ZnSO_4_): 0, 10, 100, 500, 1000, 2500, 3000, 4000, and 5000 μM. Additionally, growth was tested in 50 μM of the zinc chelating agent DTPA (diethylenetriamine pentaacetic acid, D6518-Sigma Aldrich, Saint Louis, MO, USA) with and without the supplementation of 100 μM of iron ((NH_4_)_2_Fe(SO_4_)_2_.6H_2_O) and 100 μM of copper (CuSO_4_) to check possible unspecific metal chelation by DTPA [36]. Following this test, iron and copper were added to all tested conditions. Plates were incubated at 28 °C, and growth was registered after 7 days.

For gene expression and proteomic analyses, cells were grown in Potato Dextrose Broth for 7 days at 28 °C under agitation. They were then recovered by centrifugation at 2500× *g* for 10 min, washed twice in PBS, transferred to a liquid MMcM medium, pH 6.8, containing 100 μM DTPA (zinc deprivation) and 100 μM zinc (control). Following incubation for 3 h at 150 rpm, 28 °C, the cells were collected by centrifugation and subjected to total RNA extraction, as described below. For proteomic analysis, the cells were incubated for 48 h under the same conditions, collected by centrifugation, and subjected to protein extraction (see below). In all experiments with the chemically defined medium, the glass material was treated with 10% nitric acid.

### 2.2. RNA Isolation and Gene Expression Analysis

For total RNA isolation, the cells were lysed with 0.5 μm glass beads in a mechanical beater (Mini-Beadbeater, Biospec Products Inc., Bartlesville, OK, USA) in the presence of TRIzol™ Reagent (15596026—Invitrogen™, Waltham, MA, USA) according to the manufacturer’s instructions. The samples were treated with DNase (M6101—Promega, Madison, WI, USA), followed by in vitro reverse transcription (RT) using a RevertAid RT Reverse Transcription Kit (K1691—Thermo Fisher, Waltham, MA, USA). cDNAs were submitted to a quantitative real-time PCR assay (qPCR) with a SYBR Green PCR Master Mix reagent (Applied Biosystems, Foster City, CA, USA) in a QuantStudio Real-Time PCR instrument (Applied Biosystems, Foster City, CA, USA). The sequences of the primers are shown in Appendix A. Transcript levels of the beta-tubulin gene (Z517_05482) were used in data normalization. Relative expression levels of genes were calculated using the standard curve method for relative quantification [37]. Standard curves were generated by 1/5 dilution of the cDNA solution. The data were expressed as the mean and standard deviation of experimental triplicates resulted from two independent experiments. Statistical comparisons were performed using Student’s *t*-test, and *p*-values ≤ 0.05 were considered statistically significant.

### 2.3. Protein Extraction and In-Solution Protein Digestion

To obtain a protein extract, 50 mM ammonium bicarbonate (pH 8.5) was added to cells, which were disrupted by vigorous mixing with glass beads (0.5 µm diameter) in a Mini-Beadbeater device (Mini-Beadbeater, Biospec Products Inc., Bartlesville, OK, USA). Following 8 beat cycles of 30 s intercalated with 1 min on ice, cell lysates were centrifuged at 15,000× *g* for 20 min at 4 °C until no pellet formation was obtained. The supernatant was recovered, and the protein concentration was determined by the Bradford (Sigma-Aldrich, Saint Louis, MO, USA) assay [38], with bovine serum albumin as the standard.

Samples (105 µg) were prepared for NanoUPLC-MS^E^ analysis as previously described [39]. Briefly, 52.5 µL of 0.2% (*v*/*v*) RapiGEST^TM^ (Waters Corp, Milford, MA, USA) was added to the samples and incubated at 80 °C for 15 min. The disulfide bonds were reduced by adding 2.5 µL of 100 mM DTT (Dithiothreitol—GE Healthcare, Little Chalfont, UK) and incubating at 60 °C for 30 min. Cysteine residues were alkylated by adding 2.5 µL of 300 mM iodacetamide (GE Healthcare, Piscataway, NJ, USA) and incubating in a dark at room temperature for 90 min. Subsequently, digestion was performed with 21 µL of a trypsin solution (50 ng/µL in 50 mM ammonium bicarbonate, Promega, Madison, WI, USA) at 37 °C for 16 h. Then, 21 µL of 5% (*v*/*v*) trifluoroacetic acid was added, and the mixture was incubated at 37 °C for 90 min, followed by centrifugation at 16,000× *g* at 4 °C for 30 min. The peptide mixture was dried in a speed vacuum (Eppendorf, Hamburg, Germany), followed by resuspension in 80 µL of ammonium formate (20 mM NH_4_HCO_3_, pH 10). A total of 200 fmol/µL of rabbit phosphorylase B (PHB, Waters MassPREP™ Digestion Standard) was added as the internal standard, and its signal intensity was used for absolute quantification.

### 2.4. MS Spectra Process and Proteomic Analysis

Digested peptide mixtures were analyzed by nanoscale liquid chromatography (LC, nanoACQUITY™ UPLC, Waters Corporation, Manchester, UK) coupled with tandem mass spectrometry (Synapt G1 HDMS™—Waters Corporation, Manchester, UK) operated in the MS^E^ data-independent acquisition mode. The mass spectra were detected using a [GLU1]-Fibrinopeptide B as an external (GFB) reference for mass calibration during the sample analysis and processed according to Geromanos et al. [40] using the ProteinLynx Global Server software version 3.0.2 (Waters, Manchester, UK), loaded with the *F. pedrosoi* database (https://www.uniprot.org/taxonomy/1442368 (accessed on 29 October 2021)). Parameters for protein identification consisted of: (i) the detection of at least 2 fragment ions per peptide, (ii) 5 fragments per protein, (iii) the determination of at least 1 unique (proteotypic) peptide per protein, (iv) phosphorylation of serine, threonine, and tyrosine, and oxidation of methionine were considered as variable modifications, (v) carbamidomethylation of cysteine as a fixed modification, (vi) maximum protein mass (600 kDa), (vii) one missed cleavage site was allowed for trypsin, and (viii) a false discovery rate (FDR) of 4%. Proteomic analyses were performed in triplicate.

Proteins exhibiting a 1.2-fold difference in abundance were considered to be regulated. Sequence annotation was assessed using a BlastP algorithm (http://blast.ncbi.nlm.-nih.gov/Blast.cgi (accessed on 20 May 2023)) and searches in FungiDB (https://fungidb.org/fungidb/app/ (accessed on 20 May 2023)), UniProt (https://www.uniprot.org/), and KEGG (https://www.genome.jp/kegg/ (accessed on 20 May 2023)) platforms. Functional categories were determined according to the FunCat (Functional Catalogue) classification system [41]. All online algorithms were used in default parameters.

### 2.5. Glucose and Carbohydrate Assessment

To assess glucose consumption and total carbohydrate levels, *F. pedrosoi* cells were grown under control (100 μM zinc) and zinc-limited conditions (100 μM DTPA) in biological triplicates. Glucose consumption was determined using an enzymatic system based on glucose oxidase and peroxidase reactions (MS: 10231810084, Doles^®^, Goiânia, GO, Brazil). Glucose oxidase catalyzes the oxidation of glucose, forming a red antipyrilquinonimine whose color intensity is proportional to the glucose concentration in the sample. Following incubation in zinc and DTPA conditions for 0, 16, 24, 48, and 72 h, the cultures were centrifuged at 2500× *g* for 10 min, and glucose levels were determined in the supernatants according to the manufacturer’s instructions. A standard curve was constructed using the MMcM medium with crescent glucose concentrations (0, 0.125, 0.250, 0.50, 0.75, and 1%). The absorbance measure was read at 510 nm in a SpectraMax^®^ Paradigm^®^ (Molecular Devices, Urstein, Austria).

Total carbohydrate content from *F. pedrosoi* cells was measured using the modified phenol-sulfuric acid reaction [42]. The cells were collected at a density of 0.5 g wet weight and then washed with PBS. The samples were mixed with 800 µL of concentrated sulfuric acid (98%) and 50 µL of 80% phenol aqueous solution in a microtube and read at 490 nm in a SpectraMax^®^ Paradigm^®^ (Molecular Devices, Urstein, Austria) microplate reader. A standard curve was prepared using a serial dilution of a 5 mg/mL glucose solution in water. The total carbohydrate concentration was expressed in milligrams of carbohydrate per gram of cell wet weight. The assay was conducted in triplicate, and statistical comparisons were performed using Student’s *t*-test, with *p*-values ≤ 0.01 considered statistically significant.

### 2.6. Determination of Glucan, Chitin, Neutral Lipids, and ROS

*F. pedrosoi* cells grown in control and zinc-limiting conditions for 48 h were collected by centrifugation and washed with PBS. For glucan dosage, the cells were incubated with 100% (*v*/*v*) of Aniline Blue (AB, B8563-Sigma Aldrich, Saint Louis, MO, USA) for 3 min at room temperature under stirring. Chitin levels were assessed after dyeing with 100 μg/mL of Calcofluor White (CFW, 18909-Sigma Aldrich, Saint Louis, MO, USA) for 30 min at room temperature. To assess intracellular neutral lipids, the cells were incubated with 1 µg/mL of Nile Red (NR, 72485-Sigma Aldrich, Saint Louis, MO, USA) for 15 min at room temperature. Following incubation with AB, CFW, and NR, the samples were washed twice with PBS. Intracellular reactive oxygen species (ROS) were measured using 25 nM of 2′7′-dichlorofluorescein diacetate (DCFH-DA, 35845-Sigma Aldrich, Saint Louis, MO, USA) for 7 min in the dark. AB-, CFW-, NR-, and DCFH-DA-stained suspensions were observed using an Axio-Scope A1 Microscope (Carl Zeiss AG, Oberkochen, Germany). Fluorescence intensity was determined in three biological replicas. A minimum of 50 cells per microscope slide was evaluated for each biological replicate to measure fluorescence intensity (in pixels) using AxioVision software, version 4.8.2.0 (Carl Zeiss AG, Germany). Statistical comparisons were performed using Student’s *t*-test, and *p*-values ≤ 0.01 were considered statistically significant.

### 2.7. Cell-Free ROS Measurement

*F. pedrosoi* cells grown in control and zinc-limiting conditions were collected at a density of 0.5 g wet weight and then washed with PBS. Afterwards, the cells were disrupted by vigorous mixing with glass beads (0.5 µm diameter) in a beadbeater apparatus for 5 cycles of 30 s, intercalated with 1 min on ice. Lysates were centrifuged at 15,000× *g* for 15 min at 4 °C, and the supernatant was recovered. Then, 1 µL of a 25 nM of 2′7′-dichlorofluorescein diacetate (DCFH-DA, 35845-Sigma Aldrich, Saint Louis, MO, USA) was added in 1 mL of supernatant in order to react with cell-free ROS. As a control, 2% hydrogen peroxide was utilized. Fluorescence intensity was measured with a SpectraMax^®^ Paradigm^®^ microplate reader (Molecular Devices, Urstein, Austria) at 486 nm excitation and 525 nm emission wavelengths. The fluorescence intensity values were normalized by the wet weight of cells. The assays were conducted in triplicate. Statistical comparisons were performed using Student’s *t*-test, and *p*-values ≤ 0.01 were considered statistically significant.

## 3. Results

### 3.1. F. pedrosoi Can Grow in a Wide Range of Zinc Concentrations

We have previously shown that *F. pedrosoi* has orthologous genes for zinc uptake and detoxification [34] that supposedly guarantee its survival during a nutritional challenge. However, no experimental data were available thus far to assess its ability to survive in such conditions. Therefore, we evaluated fungus growth in different zinc availabilities. As shown in Figure 1, *F. pedrosoi* could survive in a wide range of zinc concentrations (from 10 to 2500 µM) and even in conditions of zinc deprivation. Fungus growth was quite compromised in 3000 µM and completely inhibited in 5000 µM zinc. Under zinc deprivation, whether or not the chelating agent DTPA was used (DTPA and -Zn conditions), *F. pedrosoi* was able to survive. In 100 µM DTPA, a slight reduction in growth was observed. To avoid unspecific metal chelation by DTPA, MMcM was supplemented with 100 μM of iron and copper. Based on the growth results, 100 µM zinc and 100 µM DTPA were used, respectively, as control and zinc-limiting conditions in further experiments.

### 3.2. Zinc Uptake Genes Are Induced during Metal Scarcity

Genes related to zinc acquisition are generally regulated by the availability of this metal in the extracellular environment. Given the ability of *F. pedrosoi* to grow in conditions of varying zinc availability, our results suggest that the fungus has a system to maintain adequate intracellular levels of this metal. Previous bioinformatic analyses revealed the presence of zinc transporter orthologs in the *F. pedrosoi* genome, namely: the high- and low-affinity plasma membrane proteins Zrt1/ZrfB and Zrt2/ZrfA, respectively, as well as the high-affinity transporter in alkaline conditions ZrfC, and the vacuolar membrane transporter ZrcC. An ortholog of Pra1, a secreted protein that acts as an extracellular zinc capturer, was identified as well [34]. Based on these findings, gene expression analyses were carried out under zinc-limiting (100 μM DTPA) and sufficiency (100 μM Zn) conditions (Figure 2). All genes, except the vacuolar transporter *zrcC* (0.6-fold repressed), were significantly induced following 3 h of zinc deprivation. A remarkable induction was detected for the Pra1 transcript. These results lead us to conclude that these genes play a fundamental role in maintaining zinc homeostasis in the pathogenic fungus *F. pedrosoi*.

### 3.3. Overview of the Proteomic Response of F. pedrosoi to Zinc Deprivation

Considering that *F. pedrosoi* has a homeostatic machinery, represented by membrane transporters, that is activated under zinc-limiting conditions, the adaptive responses to zinc deprivation were investigated by a proteomic approach using nanoUPLC-MS^E^. The proteomic data acquired in zinc-limiting and control conditions at 48 h resulted in the identification of 530 proteins (Figure 3A). As a proteomic quality filter, a total of 493 proteins presenting the FDR cut-off (4%) and found in at least two replicates were selected for further analysis. Using a 1.2-fold cut-off, 338 proteins were found as regulated by zinc availability. Of those, 119 proteins were upregulated (Appendix A), and 219 were downregulated (Appendix A). Metabolism and protein synthesis, followed by energy, were the most enriched functional categories among the downregulated proteins. Within the upregulated group, most proteins were related to metabolism and energy (Figure 3B).

Concerning central carbon metabolism, no regulatory enzyme of glycolysis was found to be regulated, although this oxidation pathway was obviously occurring at basal levels. Citrate synthase and isocitrate dehydrogenase, two regulatory enzymes of Krebs cycle, were repressed in zinc-limiting conditions. Alcoholic fermentation was downregulated as well, as demonstrated by lower levels of both pyruvate decarboxylase and alcohol dehydrogenase. Interestingly, glucose-6-phosphate dehydrogenase, which catalyzes the first step of the pentose phosphate pathway (PPP), and transaldolase and transketolase, which are involved in the non-oxidative phase of this pathway, were upregulated. This fact led us to investigate whether glucose consumption was affected under zinc deprivation. As shown in Figure 4A, glucose concentration in the supernatant of control and DTPA cultures decreased in a similar pattern along 72 h of incubation, demonstrating that zinc-limited fungal cells were consuming this carbon source. Thus, *F. pedrosoi* captured and uses glucose, which justified the induction of the PPP.

Proteins involved in the synthesis (Z517_08984, Z517_10905) and homeostasis (Z517_01999) of pyridoxal 5′ phosphate (PLP, vitamin B6) were downregulated by zinc scarcity (Appendix A). PLP is known to act as a component of several enzymes involved in protein, lipid, and carbohydrate metabolic pathways [43] and is particularly involved in various aspects of amino acid metabolism. In agreement to the repression of PLP synthesis, a number of enzymes involved in amino acids modification as well as protein synthesis were downregulated as well.

### 3.4. Zinc Deprivation Alters Carbohydrate Levels and Distribution in F. pedrosoi

As demonstrated previously, glucose uptake was not affected in zinc-deprived cultures. The induction of enzymes related to cell wall modification, such as β-1,3-beta-glucanosyltransferase (Z517_11479) and mannosyl-oligosaccharide glucosidase (Z517_10828) raised the hypothesis that, in addition to being directed to the PPP pathway, glucose could also be directed to cell wall remodeling. To assess this, the carbohydrate content of *F. pedrosoi* was measured biochemically. Accordingly, a greater amount of total carbohydrates was observed at 72 h under zinc deprivation (Figure 4B). Glucan content at the cell surface was then measured by fluorescence microscopy (Figure 4C), which demonstrated that zinc limitation induced glucan accumulation.

Another important structural carbohydrate in the fungal cell wall is chitin, a N-acetylglucosamine (GlcNAc) homopolymer formed by β-1,4-linkages. The integrity of the fungal cell wall is maintained by crosslinks between chitin, β-glucans, and glycoproteins [44]. Staining with the fluorescent dye CFW revealed that zinc limitation triggered a reduction in the amount of chitin in the cell wall of *F. pedrosoi* (Figure 4D). Taken together, these results indicate that cell wall remodeling is an adaptive response to zinc scarcity.

### 3.5. Lipid Metabolism Is Affected by Zinc Limitation

Proteomic data demonstrated that enzymes involved in fatty acid synthesis as well oxidation were induced in zinc-limited conditions. To investigate the lipid content, NR was used to localize cytoplasmic neutral lipid droplets. As shown in Figure 5, fluorescence intensity was higher in control conditions, indicating that the quantity of neutral lipids was lower under zinc deprivation as a result of β-oxidation induction. Diphosphomevalonate decarboxylase (Z517_01009), a key enzyme in the ergosterol biosynthesis pathway, was upregulated as well. This fact, allied to the availability of acetyl-CoA, derived from β-oxidation, suggested an active synthesis of ergosterol. Zinc limitation probably induced plasma membrane remodeling together with cell wall modifications.

### 3.6. Adaptive Response to Oxidative Stress

It is known that zinc limitation triggers an increase in ROS levels as this metal is essential in the maintenance of the antioxidant defense system [45]. In agreement, several enzymes involved in ROS detoxification were induced in *F. pedrosoi* under zinc deficiency: catalase, glutathione reductase, glutathione synthetase, glutathione-S-transferase, and Fe/Mn superoxide dismutase (Appendix A). In order to verify the response of *F. pedrosoi* to oxidative stress, ROS were visualized by fluorescence microscopy following staining with DCFH-DA, a ROS marker. Fungal cells exposed to zinc limitation presented a similar pattern of intracellular ROS when compared to the control (Figure 6A,B). To reinforce this finding, ROS levels were measured after cell lysis. DCFH-DA was added to the cell-free supernatant, and the reaction was measured by spectrofluorometry. As shown in Figure 6C, the amount of ROS is similar in both zinc sufficiency and deprivation, confirming the previous analysis. These results were in agreement with the proteomic findings and suggested that the antioxidant system induced in zinc deficiency was active in managing ROS levels.

## 4. Discussion

Pathogenic fungi face metal scarcity during an infection and express high affinity uptake systems to evade the host-imposed nutritional immunity. *F. pedrosoi* is challenged and killed by NETs released by neutrophils during an infection. This host–pathogen interaction strongly suggests that this black fungus undergoes zinc starvation, since calprotectin is one of the major antifungal components of NETs [6]. We have previously shown that *F. pedrosoi* has genes related to the homeostasis of iron [35], copper, and zinc [34]. Here, growth experiments showed that this fungus could withstand zinc deprivation and maintain stable growth, as observed in fungi of the genus *Cryptococcus* [18,19], *Histoplasma* [36], and *Paracoccidioides* [46]. Fungal development was strongly affected in the medium containing 3000 µM zinc and completely inhibited by 5000 µM of the metal. Similarly, *S. cerevisiae* is able to maintain stable growth at up to 4000 μM of zinc [47]. On the other hand, *A. fumigatus* growth is severely reduced in a zinc-deficient medium, even with no addition of chelating agents [48,49]. *F. pedrosoi* growth patterns led us to investigate the fungus’ homeostatic and adaptive mechanisms to survive the stressful conditions imposed by zinc deficiency.

In a previous in silico analysis, the ZIP transporters ZrfA, ZrfB, and ZrfC were identified as part of the zinc homeostatic machinery in *F. pedrosoi*. Here, we verified that all were positively regulated by zinc deficiency, with the induction being more evident for ZrfB and ZrfC. ZrfA and ZrfB in *A. fumigatus* are both required for zinc uptake in acidic zinc-limiting conditions [49]. By contrast, *S. cerevisiae* Zrt1, a ZrfB ortholog, is upregulated in zinc limitation [50] while Zrt2, a ZrfA ortholog, is induced in the presence of zinc [51], regardless the pH. ZrfC in *A. fumigatus* and its ortholog Zrt1 in *C. albicans* are induced in neutral–alkaline zinc-limiting conditions. These transporters share the promoter region with the zincophores Aspf2 (*A. fumigatus*) and Pra1 (*C. albicans*), which present the same regulation pattern [21,52]. Interestingly, in *F. pedrosoi*, a single promoter region controls the expression of Pra1 and ZrfC/Zrt1 [34], and both genes were induced in zinc deficiency. The gene expression analyses were carried out in pH 6.8 and suggested that mainly ZrfC, Pra1, and ZrfB may be necessary for zinc uptake in such condition. Of note, all these genes present pH responsive elements (PREs) in their 5′ upstream regions, indicating that in addition to zinc availability, they are also regulated by pH [34]. Further studies are necessary to clarify the influence of pH on zinc uptake in *F. pedrosoi*. Interestingly, one plasma membrane zinc transporter was induced in the black yeast *Exophiala dermatitidis* recovered from an ex vivo skin model of infection, suggesting that zinc deprivation is experienced in this site of infection [53].

The predicted vacuolar zinc importer ZrcC was downregulated in zinc deficiency. Contrarily, the ZrcC ortholog Zrc1 in *S. cerevisiae* and *C. neoformans* is induced by zinc deprivation. In these fungi, Zrc1 is believed to protect fungal cells from zinc shock, defined as a condition when zinc abruptly enters zinc-starved cells by the high affinity transporter Zrt1 [54,55]. In *C. albicans*, ZrcC1 is not required for zinc import into the vacuole but is essential for the compartmentalization of zinc into zincosomes, storage vesicles that localizes near the vacuolar membrane [56]. Thus, ZrcC1 in *Candida* is involved in the response to variations in zinc availability, ranging from minor to toxic levels. Our results indicate that ZrcC function should not be related to the zinc shock response, but instead to maintain zinc homeostasis in zinc-sufficient conditions. The inhibition of vacuolar zinc import, suggested by the downregulation of ZrcC, results in a retention of zinc in the cytoplasm, which is necessary in zinc-limited conditions.

The regulation of membrane transporters by zinc indicated that *F. pedrosoi* promptly responds to zinc deprivation. The adaptive response to such conditions was thus investigated by a proteomic approach. Our results demonstrated that glucose uptake by *F. pedrosoi* was similar in both zinc-limited and sufficiency conditions. The assimilated glucose served as the substrate for PPP, which was induced by zinc limitation. However, the downregulation of Krebs cycle and alcoholic fermentation enzymes led us to investigate which other cellular process could be using glucose.

β-1,3-glucans, formed by glucose moieties, are the main components of fungal cell walls. The interaction of these polysaccharides with chitin and glycoproteins maintains fungal cell wall integrity. Following synthesis by β-1,3 glucan synthases, cell wall β-glucans are modified by β-1,3-glucanosyltransferases, which cleave a glycosidic linkage of a β-1,3-glucan molecule and transfer the generated residue to a non-reducing end of an acceptor β-1,3-glucan chain [57]. N-glycosylation is essential for the appropriate function of β-1,3-glucanosyltransferases. This post-translational modification is performed by a series of enzymes, including α-glucosidases. In *A. fumigatus*, N-glycosylation of β-1,3-glucanosyltransferases is essential for β-1,3-glucan synthesis [58], and activity of β-1,3-glucanosyltransferases is required for the maintenance of cell wall integrity in stress conditions [59]. Moreover, the activity of α-glucosidases is essential for virulence in *C. albicans* [60]. As β-1,3-glucanosyltransferase and mannosyl-oligosaccharide glucosidase were induced by zinc limitation, carbohydrate content in the *F. pedrosoi* cell wall was investigated. Metal scarcity induced an accumulation of β-1,3-glucans as well as a decrease in chitin content. The interaction of these polysaccharides together with glycoproteins maintain cell integrity [44], and our results suggest that cell wall remodeling is a strategy to adapt to zinc deprivation in *F. pedrosoi*. As in *F. pedrosoi*, *H. capsulatum* accumulates total carbohydrates and glycans in the cell wall [36]. Changes in cell wall organization have also been observed in the dimorphic pathogenic fungi *P. lutzii* [61] and *P. brasiliensis* [46] submitted to zinc limitation.

The cell wall and plasma membrane in fungi are in direct contact, and the mechanisms that maintain the integrity and homeostasis of both structures are intrinsically related [62]. We found that zinc limitation promoted a decrease in the neutral lipids reservoir in *F. pedrosoi*. Neutral lipids, or triacylglycerols, and sterol esters are stored in lipid droplets. These organelles provide molecules for a range of metabolic pathways in the cell, including those related to plasma membrane modifications [63]. Fungal cell membranes are composed of sphingolipids, glycerophospholipids, and sterols. Ergosterol regulates the permeability, fluidity, and integrity of the fungal plasma membrane [64]. The ergosterol biosynthetic pathway requires acetyl-CoA as a precursor and, as mentioned above, it can be stored in lipid droplets as a sterol ester [65]. Diphosphomevalonate decarboxylase, one of the enzymes involved in ergosterol synthesis, was induced in zinc-limiting conditions in *F. pedrosoi*. This fact, together with the decrease in lipid droplets and the induction of fatty acid oxidation enzymes, which generated acetyl-CoA, suggested and induction of ergosterol biosynthesis/mobilization. In *H. capsulatum* and *Mycobacterium tuberculosis*, fatty acid oxidation is activated by zinc scarcity as well [36,66].

Enzymes of the antioxidant defense system were induced by zinc limitation, suggesting that *F. pedrosoi* undergoes oxidative stress in such conditions. Indeed, many organisms experience elevated ROS levels when submitted to zinc scarcity [67], including yeast [68], fungal [69], and bacterial pathogens [66]. In *S. cerevisiae*, thioredoxin peroxidase is essential for the degradation of hydrogen peroxide [68]. In a proteomic approach, thioredoxin, glutathione reductase, and thioredoxin reductase were found to be induced by zinc limitation in *P. lutzii* [69]. The upregulation of catalase, glutathione reductase, glutathione synthetase, glutathione-S-transferase, and Fe/Mn superoxide dismutase in *F. pedrosoi* probably helped avoid an increase of ROS under zinc limitation, as demonstrated by the similar levels of these reactive species in zinc-depleted and zinc-sufficient conditions. By contrast, Cu/Zn superoxide dismutase was downregulated, which represented a strategy to save zinc for other essential pathways during metal scarcity. A survey with *S. cerevisiae* has demonstrated that in zinc-deficient cells, most of the zinc binding sites on proteins are mismetalated or unmetalated [70].

The activity of ROS detoxification enzymes depends on the availability of electrons, usually donated by NADPH. As mentioned previously, the PPP, which is the main intracellular NADPH source, was induced under zinc scarcity as well, reinforcing that *F. pedrosoi* promptly responded to ROS imbalance generated by the zinc deprivation. Interestingly, the abundance of enzymes related to sulfate assimilation decreased after exposure to the zinc chelator DTPA (Appendix A). The molecular mechanisms of sulfate assimilation are well described in yeast and filamentous fungi [71,72]. Didactically, the reactions can be subdivided into (i) the sulfate assimilation pathway, which culminates in the production of homocysteine, (ii) the transsulfuration pathway, which allows for cysteine synthesis, (iii) the methyl cycle, which includes the production of S-adenosylmethionine, and (iv) the glutathione biosynthesis pathway. Representative enzymes involved in the three first steps were downregulated under zinc scarcity. The influence of zinc status in sulfate assimilation has been described in *S. cerevisiae* [73] and *Schizosaccharomyces pombe* [74]. In these organisms, the limitation of zinc reduces the assimilation of sulfate, which is explained by the NADPH status. Zinc scarcity induces the activation of antioxidant mechanisms, which requires NADPH. The assimilation of sulfate depends on a substantial amount of NADPH as well. Thus, the reduction of sulfate assimilation is a strategy to save reducing power for ROS defense [73]. Altogether, these data from yeast and filamentous fungi suggest that decreased sulfate assimilation is an evolutionarily conserved adaptive response to zinc limitation.

## 5. Conclusions

Here we addressed the adaptive responses of *F. predrosoi* to zinc deprivation. Despite a slight reduction in growth, this dematiaceous fungus could develop in zinc-limiting conditions. Growth was supported by the induction of the homeostatic machinery of high-affinity zinc uptake, encompassed by plasma membrane transporters and the zincophore Pra1. Together with mechanisms to maintain adequate levels of zinc, *F. pedrosoi* reprograms some cellular processes in order to adapt to a zinc-poor environment. The strategies mainly include cell wall remodeling, changes in neutral lipid homeostasis, the activation of the antioxidant system, and reduction of sulfate assimilation. Collectively, such mechanisms allow *F. pedrosoi* cells to tolerate the stress caused by zinc scarcity and promote cell survival. Knowledge of the response related to the host’s nutritional immunity during an infection contributes to the definition of new therapeutic approaches. The proteomic analysis is relevant, since it reveals proteins that play an essential role in stressful conditions. The recent advance of tools for genetic manipulation in *F. pedrosoi*, which allows for the generation of knockout strains, will enable functional studies of genes related to the nutritional immunity response [75]. There is a notable increase in resistance to antimicrobial agents and the limited arsenal of antifungals. The use of zinc chelators, alone or in combination with existing drugs, has been explored as an alternative to combat fungal infections. CBM etiological agents are neglected fungi whose virulence attributes are poorly understood. Therefore, our data bring information regarding the pathobiology of *F. pedrosoi* and contribute to the 2030 Agenda of the United Nations, which highlights the significance of neglected tropical diseases.

## Figures and Tables

**Figure 1 jof-10-00118-f001:**
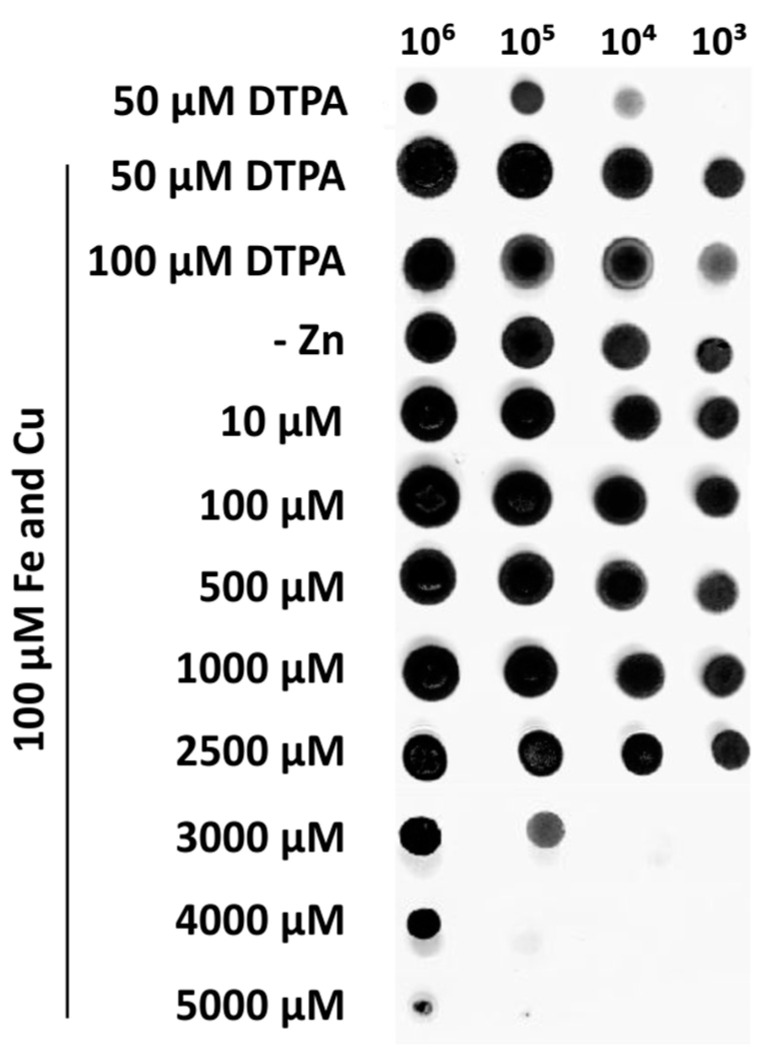
Growth of *F. pedrosoi* under conditions of different zinc availability. *F. pedrosoi* conidia were serially diluted and point-inoculated in MMcM agar plates. Growth analyses were carried out in duplicate and recorded after 7 days. *F. pedrosoi* was exposed to: 50 and 100 μM of DTPA (zinc chelator), MMcM without any addition of zinc or chelating agent (-Zn), and concentrations of 10, 100, 500, 1000, 2500, 3000, 4000, and 5000 μM of ZnSO_4_, all supplemented with 100 μM of Fe and Cu. Growth on MMcM containing 50 μM of DTPA without supplementation with Fe and Cu is also depicted.

**Figure 2 jof-10-00118-f002:**
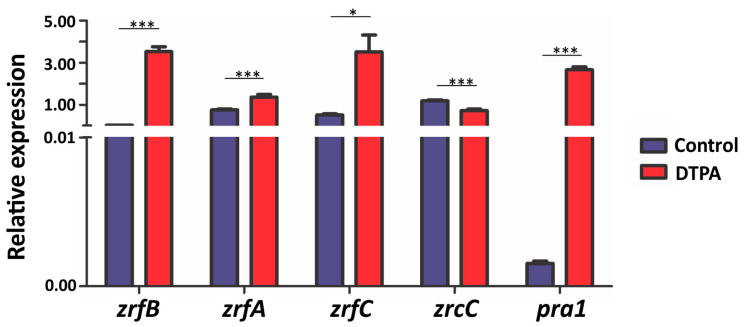
Relative expression of genes involved in zinc homeostasis in *F. pedrosoi***.** Quantitative real-time PCR was carried out with *F. pedrosoi* transcripts obtained after 3 h of exposure to 100 μM zinc (control) and 100 μM DTPA, in biological duplicates. The relative amount of mRNA was normalized using the constitutive gene encoding β-tubulin as a reference. Data are expressed as the mean ± standard deviation of experimental triplicates. Statistically significant differences were determined by Student’s *t*-test (* = *p* < 0.05; *** = *p* < 0.001).

**Figure 3 jof-10-00118-f003:**
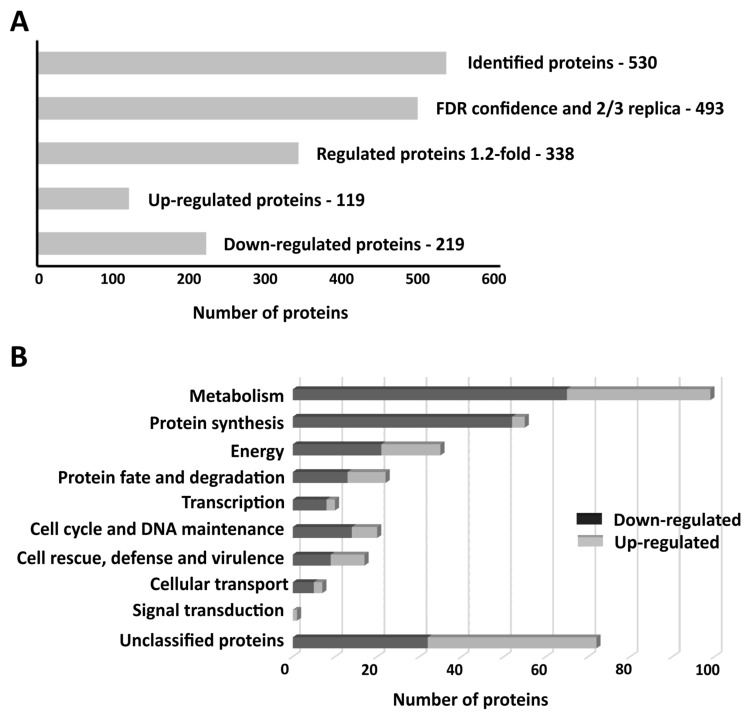
Overview of *F. pedrosoi* proteomic data generated following exposure to zinc deprivation and control conditions. (**A**) Representation of filters applied to proteins identified by the NanoUPLC-MS^E^ approach. A total of 493 proteins presented the FDR cut-off of 4% and were found in at least two of three replicates. (**B**) Functional categorization of *F. pedrosoi* proteome after 48 h under zinc-deprived conditions.

**Figure 4 jof-10-00118-f004:**
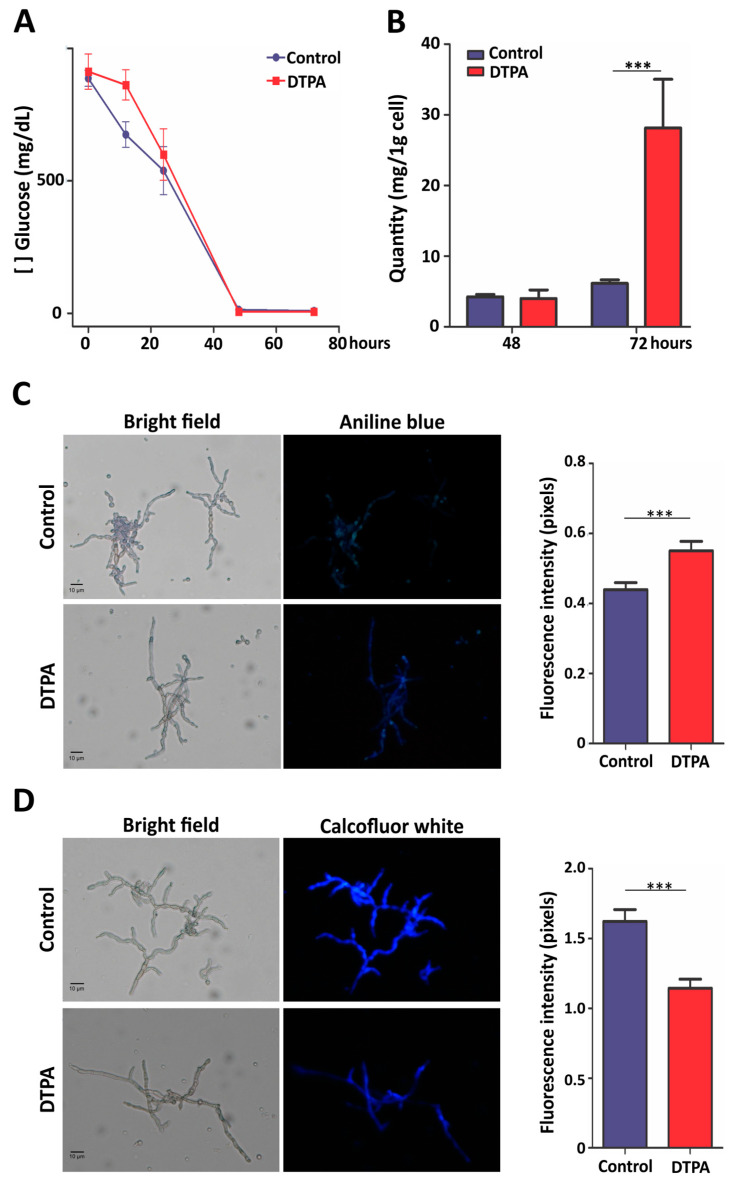
Glucose consumption and carbohydrate evaluation in *F. pedrosoi* under zinc limitation. (**A**) Glucose dosage in culture supernatants of cells grown in control and DTPA conditions. (**B**) Dosage of total carbohydrate in cells after 48 and 72 h of growth in the specified conditions. (**C**) Evaluation of β-1,3-glucans stained with Aniline Blue and visualized by fluorescence microscopy. (**D**) Detection of chitin content by Calcofluor White. The fluorescence intensity (in pixels) of the stained cells was quantified. Statistically significant differences were determined by Student’s *t*-test (*** = *p* < 0.001). Images were obtained at 400× magnification. Control: 100 μM ZnSO_4_; DTPA: 100 μM of zinc chelator DTPA.

**Figure 5 jof-10-00118-f005:**
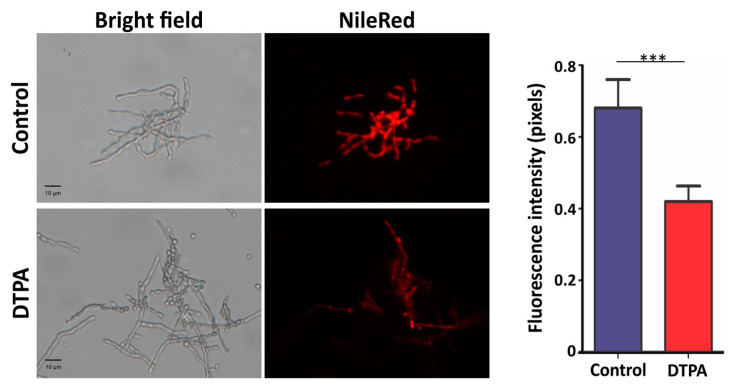
Influence of zinc limitation on lipid droplets. Lipids stained with Nile Red were visualized by fluorescence microscopy. The fluorescence intensity (in pixels) of the stained cells was quantified. Statistically significant differences were determined by Student’s *t*-test (*** = *p* < 0.001). Images were obtained at 400× magnification. Control: 100 μM ZnSO_4_; DTPA: 100 μM of zinc chelator DTPA.

**Figure 6 jof-10-00118-f006:**
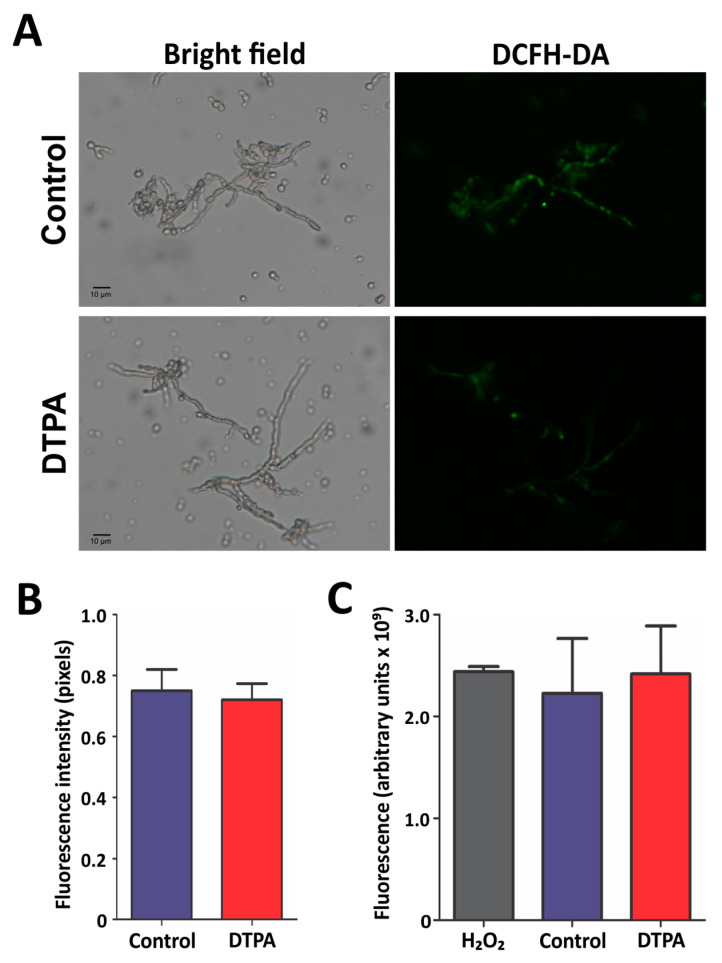
*F. pedrosoi* response to oxidative stress. (**A**) Intracellular ROS content was detected by the DCFH-DA probe and evaluated by fluorescence microscopy. (**B**) The fluorescence intensity (in pixels) of the DCFH-DA stained cells was quantified. Statistically significant differences were determined by Student’s *t*-test. The images were obtained at 400× magnification. (**C**) Cell-free ROS labeled with DCFH-DA were detected by spectrofluorometry. As a control, 2% H_2_O_2_ was used. Control: 100 μM ZnSO_4_; DTPA: 100 μM of zinc chelator DTPA.

## Data Availability

Data are contained within the article.

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
