# Peer review of "Zinc Starvation Induces Cell Wall Remodeling and Activates the Antioxidant Defense System in Fonsecaea pedrosoi"

_jof, 2024, doi:10.3390/jof10020118_

Round 1

Reviewer 1 Report

Comments and Suggestions for Authors

dear authors

I have reviewed your work, it seems to me an interesting contribution, in a well written article.

only minor corrections in the pdf file.

kind regards

Author Response

Reviewer 1

Dear authors,

I have reviewed your work, it seems to me an interesting contribution, in a well written article. Only minor corrections in the pdf file.

  • What density did you use? for what purpose?

Response: Thank you for your comment. The sentence has been replaced and clarified. See 101-102.

Reviewer 2 Report

Comments and Suggestions for Authors The article Zinc Starvation Induces Cell Wall Remodeling and Activates the Antioxidant Defense System in Fonsecaea pedrosoi Tayná by Aparecida de Oliveira Santos et al., is a very interesting preliminary work that requires greater effort to propose experimental tests that demonstrate that there is indeed an effect related to the receptors for Zn. The main objective of this work is focused on trying to demonstrate that its results are in agreement with the proteomic findings, demonstrating that the antioxidant system induced in zinc deficiency is capable of managing ROS level and unfortunately the information presented is insufficient. It would be interesting to use specific inhibitors and activators of the different stages of activation and regulation of oxidative mechanisms. These mechanisms effectively form part of the survival of pathogenic fungi in the host after invasion depends on their ability to obtain nutrients, which include the transition metal zinc. The hypothesis of this work is based on the in silico analysis of the F. pedrosoi genome previously carried out, which has shown ZIP and CDF transporters involved in the uptake and storage of zinc, although some adaptive responses to zinc deprivation are identified as a structure of the cell wall, oxidative stress state and sulfur assimilation, the work is suggested to be rewritten and an attempt is made to obtain more information that confirms the findings presented. The participation of Zn receptors should be confirmed.
Minor comments
Is not clear why most of the assays are performed using 100 micromolar only, why is not used any other divalent cation?Conclusion section is unnecessary.

Author Response

Reviewer 2

The article Zinc Starvation Induces Cell Wall Remodeling and Activates the Antioxidant Defense System in Fonsecaea pedrosoi Tayná by Aparecida de Oliveira Santos et al., is a very interesting preliminary work that requires greater effort to propose experimental tests that demonstrate that there is indeed an effect related to the receptors for Zn. The main objective of this work is focused on trying to demonstrate that its results are in agreement with the proteomic findings, demonstrating that the antioxidant system induced in zinc deficiency is capable of managing ROS level and unfortunately the information presented is insufficient. It would be interesting to use specific inhibitors and activators of the different stages of activation and regulation of oxidative mechanisms. These mechanisms effectively form part of the survival of pathogenic fungi in the host after invasion depends on their ability to obtain nutrients, which include the transition metal zinc. The hypothesis of this work is based on the in silico analysis of the F. pedrosoi genome previously carried out, which has shown ZIP and CDF transporters involved in the uptake and storage of zinc, although some adaptive responses to zinc deprivation are identified as a structure of the cell wall, oxidative stress state and sulfur assimilation, the work is suggested to be rewritten and an attempt is made to obtain more information that confirms the findings presented. The participation of Zn receptors should be confirmed.

Minor comments

Is not clear why most of the assays are performed using 100 micromolar only, why is not used any other divalent cation?Conclusion section is unnecessary.

Response:

Dear reviewer, thanks for your comments on our manuscript. The hypothesis of this work is that Fonsecaea pedrosoi may undergo zinc starvation along the infection. It was based on previous findings that fungal cells are killed by NETs, which contain the zinc chelator calprotectin, released by neutrophils at the site of infection. According to the hypothesis, the main goal of this work was to investigate the responses of F. pedrosoi to zinc deprivation at a molecular level. For that, we employed gene expression analysis and proteomics as the main approaches. In a previous in silico work, we defined ZIP and CDF transporters that could be involved in zinc homeostasis in F. pedrosoi, based on sequence homology. In the present study, we investigated if those potentially involved in high affinity zinc uptake (zrfB, zrfA, zrfC and pra1) were induced by zinc limitation. As all of them were significantly upregulated, we concluded that they play a role in zinc acquisition and that the fungus was really submitted to a zinc starved condition. Functional studies (as the generation of knockout strains) are necessary to define the detailed function of each transporter (that I believe the reviewer named as “Zn receptors”) but are beyond the objectives of this article. Concerning the management of ROS in zinc-limiting conditions, we performed two assays that demonstrated ROS levels in both zinc sufficiency and zinc-limiting conditions: fluorescence microscopy (where ROS are observed in intact cells) and ROS measurement after cell lysis. In both conditions, ROS levels were not significantly different. As various enzymes of the antioxidant system were induced in zinc limitation, we suggest that their activity is contributing to manage ROS levels in such condition. As this is the first report of F. pedrosoi response to zinc limitation, our goal is to give a broader view of fungus metabolism under this stressful condition. We thank the reviewer for the suggestion to deepen the investigation regarding the oxidative mechanisms, but it is beyond the scope of our manuscript and can be a subject of further investigations. Anyway, we changed some terms in the manuscript (lines 367-369, 472-475), as demonstrated bellow, in capital letters:

“These results are in agreement to the proteomic findings and SUGGESTS that the antioxidant system induced in zinc deficiency is active in managing ROS levels.”

“The upregulation of catalase, glutathione reductase, glutathione synthetase, glutathione-S-transferase and Fe/Mn superoxide dismutase in F. pedrosoi PROBABLY avoided an increase of ROS under zinc limitation, as demonstrated by the similar levels of these reactive species in zinc depleted and zinc sufficient conditions”

Regarding the minor comments:

It is not clear if “100 micromolar” refers to DTPA concentration. If so, 100 uM DTPA was used as the condition of zinc deprivation based on: (i) the growth experiments, in which a slight reduction in growth was observed in 100 uM DTPA in comparison to the conditions where zinc was added in different concentrations and (ii) the gene expression analysis, which demonstrated the induction of genes related to the high affinity zinc uptake. DTPA is an extracellular metal chelating agent that have been successfully used to mimic zinc deprived conditions in investigations with other pathogenic fungi, namely Cryptococcus neorformans (Schneider et al, 2015) and Histoplasma capsulatum (Assunção et al, 2020). In both studies, DTPA was used at a concentration of 100 uM.

Schneider RO, Diehl C, Dos Santos FM, Piffer AC, Garcia AWA, Kulmann MIR, Schrank A, Kmetzsch L, Vainstein MH, Staats CC. Effects of zinc transporters on Cryptococcus gattii virulence. Sci Rep. 2015 May 7;5:10104. doi: 10.1038/srep10104. PMID: 25951314; PMCID: PMC4423424.

Assunção LDP, Moraes D, Soares LW, Silva-Bailão MG, de Siqueira JG, Baeza LC, Báo SN, Soares CMA, Bailão AM. Insights Into Histoplasma capsulatum Behavior on Zinc Deprivation. Front Cell Infect Microbiol. 2020 Nov 30;10:573097. doi: 10.3389/fcimb.2020.573097. PMID: 33330123; PMCID: PMC7734293.

Reviewer 3 Report

Comments and Suggestions for Authors

Dear authors,

I read your original article concerning how zinc Starvation Induces Cell Wall Remodeling and Activates the Antioxidant Defense System in Fonsecaea pedrosoi. The pape ris well-written, easy to read and present innovative results starting from previous analyses, both in silico and in vitro. I appreciqted this translational research. Figures are adeguate and language is fine. I report some points to address.

1)      Abstract line 16, Replace Scarcity with zinc Zn-limiting conditions.

2)      In the introduction, you report an interesting point about psoriasin. It should be noted that in psoriasin acts against pathogens. This is true but it should be more appropriate to refer to S100 family. Moreover, regarding psoriasis, it should be reported that in parallel with zinc, other micronutrients as vitamin A and its derivates, play a role in defence against fungal infections. I suggest reading and citing :

-      Campione E, Cosio T, Lanna C, Mazzilli S, Ventura A, Dika E, Gaziano R, Dattola A, Candi E, Bianchi L. Predictive role of vitamin A serum concentration in psoriatic patients treated with IL-17 inhibitors to prevent skin and systemic fungal infections. J Pharmacol Sci. 2020 Sep;144(1):52-56. doi: 10.1016/j.jphs.2020.06.003. Epub 2020 Jun 11. PMID: 32565006

3)      « To avoid unspecific metal chelation by DTPA, 100 μM of iron ((NH4)2Fe(SO4)2.6H2O) and 99 100 μM of copper (CuSO4) was added to all tested conditions. » A single experiment with only DTPA should be performed or reference must be reported here.

4)      I suggest reading the following articles to improve discussion and introduction :

-      Favilla LD, Herman TS, Goersch CDS, de Andrade RV, Felipe MSS, Bocca AL, Fernandes L. Expanding the Toolbox for Functional Genomics in Fonsecaea pedrosoi: The Use of Split-Marker and Biolistic Transformation for Inactivation of Tryptophan Synthase (trpB) Gene. J Fungi (Basel). 2023 Feb 8;9(2):224. doi: 10.3390/jof9020224. PMID: 36836338; PMCID: PMC9963410.

-      Guevara A, Siqueira NP, Nery AF, Cavalcante LRDS, Hagen F, Hahn RC. Chromoblastomycosis in Latin America and the Caribbean: Epidemiology over the past 50 years. Med Mycol. 2021 Dec 8;60(1):myab062. doi: 10.1093/mmy/myab062. PMID: 34637525.

-      Mourad AI, Haber RM. Visual Dermatology: Extensive Chromoblastomycosis of the Leg Secondary to Fonsecaea pedrosoi. J Cutan Med Surg. 2022 Jan-Feb;26(1):101. doi: 10.1177/1203475420960434. Epub 2020 Sep 18. PMID: 32945204.

-      Collopy-Junior I, Kneipp LF, da Silva FC, Rodrigues ML, Alviano CS, Meyer-Fernandes JR. Characterization of an ecto-ATPase activity in Fonsecaea pedrosoi. Arch Microbiol. 2006 Jun;185(5):355-62. doi: 10.1007/s00203-006-0100-1. Epub 2006 Mar 10. PMID: 16528535.

-      Espinel-Ingroff A, Goldson PR, McGinnis MR, Kerkering TM. Evaluation of proteolytic activity to differentiate some dematiaceous fungi. J Clin Microbiol. 1988 Feb;26(2):301-7. doi: 10.1128/jcm.26.2.301-307.1988. PMID: 3343325; PMCID: PMC266272.

Comments on the Quality of English Language

Minor editing of English language required

Author Response

Reviewer 3

Dear authors,

I read your original article concerning how zinc Starvation Induces Cell Wall Remodeling and Activates the Antioxidant Defense System in Fonsecaea pedrosoi. The paper is well-written, easy to read and present innovative results starting from previous analyses, both in silico and in vitro. I apprecieted this translational research. Figures are adeguate and language is fine. I report some points to address.

  • Abstract line 16, Replace Scarcity with zinc Zn-limiting conditions.

Response: The term was changed, as required (see line 16).

  • In the introduction, you report an interesting point about psoriasin. It should be noted that in psoriasin acts against pathogens. This is true but it should be more appropriate to refer to S100 family. Moreover, regarding psoriasis, it should be reported that in parallel with zinc, other micronutrients as vitamin A and its derivates, play a role in defence against fungal infections. I suggest reading and citing :

-      Campione E, Cosio T, Lanna C, Mazzilli S, Ventura A, Dika E, Gaziano R, Dattola A, Candi E, Bianchi L. Predictive role of vitamin A serum concentration in psoriatic patients treated with IL-17 inhibitors to prevent skin and systemic fungal infections. J Pharmacol Sci. 2020 Sep;144(1):52-56. doi: 10.1016/j.jphs.2020.06.003. Epub 2020 Jun 11. PMID: 32565006

Response: Thank you for your assistance. The article was cited in the Introduction section (lines 35-38)

  • « To avoid unspecific metal chelation by DTPA, 100 μM of iron ((NH4)2Fe(SO4)2.6H2O) and 100 μM of copper (CuSO4) was added to all tested conditions. » A single experiment with only DTPA should be performed or reference must be reported here.

Response: Thank you for your observation. The growth test had already been performed and the result were added to Figure 1. Additionally, a reference was reported. The information can be found in lines 104-108, 241-243 and 251-252.

  • I suggest reading the following articles to improve discussion and introduction :

-  Favilla LD, Herman TS, Goersch CDS, de Andrade RV, Felipe MSS, Bocca AL, Fernandes L. Expanding the Toolbox for Functional Genomics in Fonsecaea pedrosoi: The Use of Split-Marker and Biolistic Transformation for Inactivation of Tryptophan Synthase (trpB) Gene. J Fungi (Basel). 2023 Feb 8;9(2):224. doi: 10.3390/jof9020224. PMID: 36836338; PMCID: PMC9963410.

-  Guevara A, Siqueira NP, Nery AF, Cavalcante LRDS, Hagen F, Hahn RC. Chromoblastomycosis in Latin America and the Caribbean: Epidemiology over the past 50 years. Med Mycol. 2021 Dec 8;60(1):myab062. doi: 10.1093/mmy/myab062. PMID: 34637525.

- Mourad AI, Haber RM. Visual Dermatology: Extensive Chromoblastomycosis of the Leg Secondary to Fonsecaea pedrosoi. J Cutan Med Surg. 2022 Jan-Feb;26(1):101. doi: 10.1177/1203475420960434. Epub 2020 Sep 18. PMID: 32945204.

-      Collopy-Junior I, Kneipp LF, da Silva FC, Rodrigues ML, Alviano CS, Meyer-Fernandes JR. Characterization of an ecto-ATPase activity in Fonsecaea pedrosoi. Arch Microbiol. 2006 Jun;185(5):355-62. doi: 10.1007/s00203-006-0100-1. Epub 2006 Mar 10. PMID: 16528535.

-      Espinel-Ingroff A, Goldson PR, McGinnis MR, Kerkering TM. Evaluation of proteolytic activity to differentiate some dematiaceous fungi. J Clin Microbiol. 1988 Feb;26(2):301-7. doi: 10.1128/jcm.26.2.301-307.1988. PMID: 3343325; PMCID: PMC266272.

Response: Thank you for the observation. Most of the suggested articles were cited in the Introduction and Conclusions. Additional information regarding the disease was added was well (lines 71, 73-75, 78-79, 512-515).

  • Comments on the Quality of English Language. Minor editing of English language required.

Response: Thank you for your feedback. The authors have revised the text in response to your comment.

Round 2

Reviewer 2 Report

Comments and Suggestions for Authors

my main queries were satisfied